REGISTERED REPORT

# Registered report: Inhibition of BET recruitment to chromatin as an effective treatment for MLL-fusion leukemia

**Juan José Fung[1], Alan Kosaka[1], Xiaochuan Shan[2], Gwenn Danet-Desnoyers[2], Michael Gormally[3], Kate Owen[4], Reproducibility Project: Cancer Biology***

[1]ProNovus Bioscience, Mountain View, California; [2]Stem Cell and Xenograft Core, University of Pennsylvania, Perelman School of Medicine, Philadelphia, Pennsylvania; [3]University of Cambridge, Cambridge, United Kingdom; [4]University of Virginia, Charlottesville, Virginia

## REPRODUCIBILITY PROJECT CANCER BIOLOGY

**Abstract** The Reproducibility Project: Cancer Biology seeks to address growing concerns about reproducibility in scientific research by conducting replications of selected experiments from a number of high-profile papers in the field of cancer biology. The papers, which were published between 2010 and 2012, were selected on the basis of citations and Altmetric scores (*Errington et al., 2014*). This Registered report describes the proposed replication plan of key experiments from 'Inhibition of bromodomain and extra terminal (BET) recruitment to chromatin as an effective treatment for mixed-lineage leukemia (MLL)-fusion leukemia' by Dawson and colleagues, published in *Nature* in 2011 (*Dawson et al., 2011*). The experiments to be replicated are those reported in Figures 2A, 3D, 4B, 4D and Supplementary Figures 11A-B and 16A. In this study, BET proteins were demonstrated as potential therapeutic targets for modulating aberrant gene expression programs associated with MLL-fusion leukemia. In Figure 2A, the BET bromodomain inhibitor I-BET151 was reported to suppress growth of cells harboring MLL-fusions compared to those with alternate oncogenic drivers. In Figure 3D, treatment of MLL-fusion leukemia cells with I-BET151 resulted in transcriptional suppression of the anti-apoptotic gene *BCL2*. Figures 4B and 4D tested the therapeutic efficacy of I-BET151 in vivo using mice injected with human MLL-fusion leukemia cells and evaluated disease progression following I-BET151 treatment. The Reproducibility Project: Cancer Biology is a collaboration between the Center for Open Science and Science Exchange and the results of the replications will be published in *eLife*.

*For correspondence: tim@cos.io

Group author details
Reproducibility Project: Cancer Biology
See page 17

## Introduction

The mixed-lineage leukemia (*MLL*) gene encodes a large histone methyltransferase that directly binds DNA and positively regulates gene transcription (*Marschalek, 2010*). *MLL* is a frequent target of chromosomal translocation events (*Meyer et al., 2009*). During rearrangement, the N-terminus of *MLL* fuses to one of more than 60 partners, the most common of which coexist in a super elongation complex (SEC) enriched with transcription elongation factors (*Meyer et al., 2009*; *Smith et al., 2011*). The resulting fusion event converts *MLL* into a potent transcriptional activator often giving rise to aggressive hematological malignancies (*Mueller et al., 2009*; *Slany, 2009*). The overall prognosis for pediatric and adult patients with confirmed MLL-fusion leukemia remains extremely poor and necessitates the development of new methodologies and therapeutic agents to improve survival outcomes (*Slany, 2009*; *Tamai and Inokuchi, 2010*).

Bromodomain and extra terminal (BET) proteins are transcriptional regulators that epigenetically control the expression of genes involved in cell cycle, growth and inflammation (*Darnell, 2002*; *Wu and Chiang, 2007*; *LeRoy et al., 2008*; *Dey et al., 2009*; *Nicodeme et al., 2010*). BETs therefore

provide potential therapeutic targets for modulating gene expression programs associated with various human diseases. Dawson and colleagues identified novel interactions between BET family members bromodomain protein (BRD) 3 and BRD4 with components of the SEC and polymerase-associated factor complexes in MLL fusion cells (*Dawson et al., 2011*). Given that BRD3 and BRD4 may be involved in the recruitment of the SEC and PAF complexes to regions of active chromatin, the authors tested the hypothesis that the dislocation of BET proteins from chromatin constitutes a viable therapeutic strategy in the treatment of MLL-fusion leukemia. For this purpose, Dawson and colleagues developed I-BET151, a BET inhibitor that selectively binds to the bromodomains of BET proteins and prevents their ability to bind acetylated histone residues (*Dawson et al., 2011*).

In Figure 2A and S11A-B, Dawson and colleagues assessed the ability of I-BET151 to suppress cell growth in a variety of human leukemia cell lines (*Dawson et al., 2011*). In these experiments, cells were treated with increasing concentrations of I-BET151 and allowed to grow for a further 72 hr. I-BET151 treatment was extremely effective at inhibiting the growth of leukemic cell lines harboring MLL fusions, including MV4;11, RS4;11, MOLM13, and NOMO1 cells, as determined by their low (nanomolar range) $IC_{50}$ values. In contrast, the proliferation of cell lines using other oncogenic drivers, including gain-of-function kinase activity, was either resistant (K526) or significantly less sensitive (human erythroleukemic [HEL], HL60, and MEG01 cells) to I-BET151, exhibiting $IC_{50}$ concentrations in the micromolar range and above. This key experiment shows that I-BET151 exhibits potent efficacy against MLL-fusion leukemic cell lines and will be replicated in Protocol 2. More recently, substantial growth inhibition with I-BET151 has been observed in other hematological malignancies, including acute myeloid leukemia (AML) (*Dawson et al., 2014*), multiple myeloma (MM) (*Chaidos et al., 2014*), and primary effusion lymphoma (*Tolani et al., 2014*), as well as non-hematological cancer models (medulloblastoma, melanoma, and glioblastoma) at concentrations ranging from 100 to 500 nM (*Gallagher et al., 2014*; *Long et al., 2014*; *Pastori et al., 2014*). Additionally, the BET inhibitor JQ1 was reported to have a broad growth-suppressive activity, similar to I-BET151, effectively inhibiting leukemic cell lines, such as MV4;11, while K526 cells remained largely resistant (*Zuber et al., 2011*).

To investigate the mechanism of action for I-BET151, Dawson and colleagues assessed apoptosis and cell cycle progression after drug treatment. Closer examination of the transcriptional pathways controlled by I-BET151 revealed that drug treatment repressed the activity of several known *MLL* targets, including the oncogene *BCL-2*. Bcl-2 promotes cell survival and protects cells from a wide range of cytotoxic insults (*Cory et al., 2003*). In Figure 3D, the authors confirmed the ability of I-BET151 to transcriptionally downregulate *BCL-2* expression in the MLL-fusion cell lines MOLM13, MV4;11, and NOMO1, but not in the K526 resistant cell line. This key experiment shows that I-BET151 is effective at silencing *BCL-2* gene transcription and will be replicated in Protocol 3. In addition to MLL-fusion cell lines, I-BET151 treatment correlated with enhanced apoptosis and reduced *BCL-2* gene transcription in AML patient samples (*Dawson et al., 2014*). In contrast, while I-BET151 also promoted cell death and/or growth inhibition in HEL cells (*Wyspianska et al., 2014*), Me1007 melanoma cells (*Gallagher et al., 2014*), and Sufu$^{-/-}$ cells (mouse embryo fibroblasts deficient in the hedge hog signaling molecule Smoothened) (*Long et al., 2014*), drug treatment did not significantly impact Bcl-2 at either the gene or protein expression level.

In Figure 4B and 4D (and Supp. Figure 16A), the therapeutic potential of I-BET151 treatment was tested in vivo. Using a well-established model of disseminated MLL leukemia, animals were treated with I-BET151 21 days after transplantation with MV4;11 cells and monitored for clinical signs of disease. Here, Dawson and colleagues showed that I-BET151 significantly improved the length of disease-free survival and reduced evidence of peripheral blood (PB) disease compared to vehicle-treated animals. Similar findings recapitulating the suppressive effect of I-BET151 on tumor growth have been reported in medulloblastoma, melanoma, and glioblastoma xenograft models (*Gallagher et al., 2014*; *Long et al., 2014*; *Pastori et al., 2014*). Similarly, follow-up studies by Dawson and colleagues demonstrated that I-BET151 confers a significant survival advantage and reduces the circulating leukemic burden in a murine model of AML (*Dawson et al., 2014*). These experiments will be replicated in Protocols 4 and 5. Similar studies testing the efficacy of JQ1, an independent BET inhibitor, reported a decrease in tumor growth in nude mice bearing AML xenografts (MV4;11 cells) (*Mertz et al., 2011*) and SCID-beige mice bearing MM xenografts (MM.1S cells) (*Delmore et al., 2011*).

# Materials and methods

## Protocol 1: Determine the population doubling time of K562 and MV4;11 cells

The doubling time of K-562 and MV4;11 cells is assumed to be approximately 25 and 50 hr, respectively. To empirically determine the doubling time in the replicating lab, this general protocol will be used to determine the treatment time of K562 and MV4;11 cells for Protocol 2.

### Sampling

- This experiment is performed with two cell lines (K562 and MV4;11).
- Each cell line to be performed with six technical repeats per experiment.
- The experiment is performed a total of once.

### Materials and reagents

| Reagent | Type | Manufacturer | Catalog # | Comments |
| --- | --- | --- | --- | --- |
| MV4;11 | Human cell line | ATCC | CRL-9591 | – |
| K-562 | Human cell line | ATCC | CCL-243 | – |
| RPMI-1640 medium, with L-glutamine and sodium bicarbonate | Cell culture reagent | Sigma–Aldrich | R8758 | Original catalog number not specified |
| Fetal bovine serum (FBS) | Cell culture reagent | Sigma–Aldrich | F2442 | Original brand not specified |
| Penicillin–streptomycin solution (100x) stabilized | Cell culture reagent | Sigma–Aldrich | P4333 | Original brand not specified |
| T-75 flasks | Labware | Corning | 430641U | Original brand not specified |
| 96-well tissue culture plates (optically clear) | Labware | Corning | 3595 | Original brand not specified |
| Cell-titer aqueous one solution cell proliferation assay (MTS) | Assay kit | Promega | G3582 | – |
| Plate reader capable of reading absorbance at 490 nm | Instrument | Molecular Devices | SpectraMax 190 | Replaces Gemini reader |
| Softmax Pro | Software | Molecular Devices | Version 3.1.2 | – |

### Procedure

#### Note

- All cells will be sent for mycoplasma testing and STR profiling.
- MV4;11 and K-562 human leukemic cells are maintained in RPMI-1640 medium supplemented with 10% fetal bovine serum (FBS) and 1% penicillin/streptomycin at 37°C with 5% $CO_2$.

1. Seed between $4 \times 10^4$ and $1 \times 10^5$ cells into two 96-well tissue culture plates with 100 µl of medium per well, excluding outer wells. Incubate cells overnight at 37°C with 5% $CO_2$.
   a. Fill outer wells with medium alone.
   b. Include six non-outer wells with medium alone for background subtraction.
2. With one plate perform MTS Assay (Promega CellTiter-Aqueous One) according to manufacturer's instructions.
   a. Incubate plates for 4 hr at 37°C.
   b. Read absorbance at 490 nm.
   c. Subtract background (average of medium only wells) from wells with cells and determine average reading from first plate.
3. 3 days later perform MTS Assay (Promega CellTiter-Aqueous One) according to manufacturer's instructions on second plate.
   a. Incubate plates for 4 hr at 37°C.
   b. Read absorbance at 490 nm.

 c. Subtract background (average of medium only wells) from wells with cells and determine average reading from second plate.
4. Calculate the population doubling time for each cell line using the following formula:
 a. Doubling time = ln2/ln(second plate average reading/first plate average reading).

## Deliverables

- Data to be collected:
  ○ STR profile and result of mycoplasma testing.
  ○ Raw data and background subtracted absorbance at 490 nm.

## Confirmatory analysis plan

- n/a.

## Known differences from the original study

All known differences are listed in the materials and reagents section above with the originally used item listed in the comments section. All differences have the same capabilities as the original and are not expected to alter the experimental design.

## Provisions for quality control

The cell lines used in this experiment will undergo STR profiling to confirm their identity and will be sent for mycoplasma testing to ensure there is no contamination. All of the raw data will be uploaded to the project page on the OSF (https://osf.io/hcqqy) and made publically available.

## Protocol 2: Cell viability assay to determine selective inhibition of an MLL-fusion leukemic cell line with I-BET151

This protocol assesses the ability of I-BET151, a small molecule inhibitor of BET family proteins, to selectively and potently inhibit the growth of the human leukemic cell line MV4;11, which is driven by an oncogenic translocation of the *MLL* gene. As a negative control, human K-562 leukemic cells, which are not oncogenically driven by an MLL-fusion, will also be treated with I-BET151. As a further negative control, both cell lines will be treated with vehicle alone (dimethyl sulfoxide (DMSO)). This protocol will replicate experiments reported in Figure 2A, Supp. Figure 11A, and Supp. Figure 11B.

### Sampling

- This experiment will be performed three separate times (biological replicates) for a final power of $\geq$80%. The original data reported a single $IC_{50}$ value for each cell line, thus to determine an appropriate number of replicates to perform initially, sample sizes required based on a range of potential variance was determined. The sample size will also be determined *post hoc* as described in 'Power calculations' and additional replicates will be performed if necessary.
  ○ See 'Power calculations' section for details.
- Experiment has two cohorts:
  ○ K562 human leukemic cells (−MLL).
  ○ MV4;11 human leukemic cells (+MLL).
- Each cohort has 11 conditions to be performed in technical triplicate per experiment:
  ○ DMSO (vehicle).
  ○ 0.01 nM I-BET151.
  ○ 0.1 nM I-BET151.
  ○ 1 nM I-BET151.
  ○ 10 nM I-BET151.
  ○ 100 nM I-BET151.
  ○ 1 μM I-BET151.
  ○ 10 μM I-BET151.
  ○ 100 μM I-BET151.
  ○ 1 mM I-BET151.
  ○ 10 mM I-BET151.

## Materials and reagents

| Reagent | Type | Manufacturer | Catalog # | Comments |
|---|---|---|---|---|
| MV4;11 | Human cell line | ATCC | CRL-9591 | – |
| K-562 | Human cell line | ATCC | CCL-243 | – |
| I-BET151 (GSK1210151A) | Small molecule | Sigma–Aldrich | SML0666 | – |
| RPMI-1640 medium, with L-glutamine and sodium bicarbonate | Cell culture reagent | Sigma–Aldrich | R8758 | Original catalog number not specified |
| Fetal bovine serum (FBS) | Cell culture reagent | Sigma–Aldrich | F2442 | Original brand not specified |
| Penicillin–streptomycin solution (100×) stabilized | Cell culture reagent | Sigma–Aldrich | P4333 | Original brand not specified |
| T-75 flasks | Labware | Corning | 430641U | Original brand not specified |
| 96-well tissue culture plates (optically clear) | Labware | Corning | 3595 | Original brand not specified |
| 96-well sterile plate (for preparing compound dilutions) | Labware | Corning | 3370 | Original brand not specified |
| DMSO, molecular biology grade | Reagent | Sigma–Aldrich | D8418 | Original brand not specified |
| Cell-titer aqueous one solution cell proliferation assay (MTS) | Assay kit | Promega | G3582 | – |
| Plate reader capable of reading absorbance at 490 nm | Instrument | Molecular Devices | SpectraMax 190 | Replaces Gemini reader |
| Softmax Pro | Software | Molecular Devices | Version 3.1.2 | – |

## Procedure

### Note

- All cells will be sent for mycoplasma testing and STR profiling.
- MV4;11 and K-562 human leukemic cells maintained in RPMI-1640 medium, supplemented with 10% FBS and 1% penicillin/streptomycin at 37°C with 5% $CO_2$.

1. Seed between $4 \times 10^4$ and $1 \times 10^5$ cells into 96-well tissue culture plates with 90 µl of medium per well, excluding outer wells. Incubate cells overnight at 37°C with 5% $CO_2$.
   a. Fill outer wells with medium alone.
   b. Include at least three non-outer wells with medium alone for background subtraction.
   c. One plate for each cell line with 33 wells seeded with cells for each plate.
2. Treat cells with 10 µl of 10× serial dilutions of I-BET151 to yield final dilutions of 0.01 nM–10 mM (10 dilutions), or treat with DMSO (vehicle) control.
   a. Dilute stock of I-BET151 at 1000× final concentration of serial dilution stocks in DMSO (10 nM–10 M).
   b. Dilute 1000× serial dilution stocks 1:100 in complete growth medium to yield a 10× stock (0.1 nM–100 mM) that is added directly to the 90 µl of cell/medium.
      i. Final DMSO concentration kept to 0.1% DMSO.
3. Incubate cells for approximately three times the doubling time of each cell line.
   a. The doubling time for each cell line is determined in Protocol 1.
4. Perform MTS Assay (Promega CellTiter-Aqueous One®) according to manufacturer's instructions.
   a. Incubate plates for 4 hr at 37°C.
   b. Read absorbance at 490 nm.
   c. Calculate viability as a percentage of control (DMSO (vehicle) cells) after background subtraction.
5. Determine IC50 values for each cell line.
6. Repeat independently two additional times.

### Deliverables

- Data to be collected:
   ○ STR profile and result of mycoplasma testing of cells.

○ Raw absorbance data, I-BET151 values at each concentration normalized to DMSO-treated control values, and analyzed data (sigmoidal dose–response curves for I-BET151), and $IC_{50}$ values determined for each cell line and repeat. (Compare to Figures S11A and S11B).

## Confirmatory analysis plan

- Statistical analysis of the replication data:
  ○ Unpaired two-tailed $t$-test of the $IC_{50}$ values for I-BET151 treated K-562 cells will be compared to $IC_{50}$ values for I-BET151 treated MV4;11 cells.
- Meta-analysis of original and replication attempt effect sizes:
  ○ The replication data (mean and 95% confidence interval) will be plotted with the original reported data value displayed as a single point on the same plot for comparison.

## Known differences from the original study

All known differences are listed in the materials and reagents section above with the originally used item listed in the comments section. All differences have the same capabilities as the original and are not expected to alter the experimental design.

## Provisions for quality control

The cell lines used in this experiment will undergo STR profiling to confirm their identity and will be sent for mycoplasma testing to ensure there is no contamination. The doubling time of each cell line was determined in Protocol 1. All of the raw data will be uploaded to the project page on the OSF (https://osf.io/hcqqy) and made publically available.

## Protocol 3: qPCR analysis of *BCL2* gene expression following I-BET151 treatment

This protocol evaluates the expression of the *BCL2* gene in both MV4;11 (+MLL) and K-562 (−MLL) leukemic cell lines following treatment with the BET inhibitor I-BET151. *BCL2* is a key anti-apoptotic gene implicated in the pathogenesis of MLL-fusion leukemias. Treatment with I-BET151 is expected to reduce the expression of *BCL2* in MV4;11 cells, but not in the unresponsive K-562 cells. As a control, both cell lines will also be treated with vehicle alone (DMSO only). The expression of *BCL2* will be normalized against the endogenous expression of $\beta_2$ microglobulin (*B2M*). This protocol will replicate experiments reported in Figure 3D.

### Sampling

- Perform this experiment three separate times (biological replicates) for a minimum power of 80%.
  ○ See 'Power calculations' section for details.
- Experiment has two cohorts:
  ○ K562 human leukemic cells (−MLL).
  ○ MV4;11 human leukemic cells (+MLL).
- Each cohort has two conditions to be performed in technical duplicate per experiment (qRT-PCR of *BCL2* and *B2M*):
  ○ DMSO (vehicle).
  ○ 500 nM I-BET151.

### Materials and reagents

| Reagent | Type | Manufacturer | Catalog # | Comments |
|---|---|---|---|---|
| MV4;11 | Human cell line | ATCC | CRL-9591 | – |
| K-562 | Human cell line | ATCC | CCL-243 | – |
| I-BET151 (GSK1210151A) | Small molecule | Sigma–Aldrich | SML0666 | – |
| DMSO, molecular biology grade | Reagent | Sigma–Aldrich | D8418 | Original brand not specified |
| RPMI-1640 medium, with L-glutamine and sodium bicarbonate | Cell culture reagent | Sigma–Aldrich | R8758 | Original catalog number not specified |

*Continued on next page*

*Continued*

| Reagent | Type | Manufacturer | Catalog # | Comments |
|---------|------|--------------|-----------|----------|
| Fetal bovine serum (FBS) | Cell culture reagent | Sigma–Aldrich | F2442 | Original brand not specified |
| Penicillin–streptomycin solution (100×) stabilized | Cell culture reagent | Sigma–Aldrich | P4333 | Original brand not specified |
| 48-well tissue culture plates | Labware | Corning | 3548 | Original brand not specified |
| RNAspin mini | RNA isolation | Sigma–Aldrich | GE25-0500-70 | Replaces Qiagen cat. no. 74104 used in original study |
| Nuclease-free water (DEPC-treated) | Chemical | Sigma–Aldrich | 95284 | Reagent needed for RNAspin Mini protocol |
| 96-well plates (for quantification of RNA) | Labware | Corning | 3635 | UV/Vis 96-well clear plates for use on Molecular Devices Spectramax 190 |
| First-strand cDNA synthesis kit | cDNA synthesis | Sigma–Aldrich | GE27-9261-01 | Replaces Invitrogen cat. no. 28025-013 used in original study |
| *BCL2*-primers (forward and reverse) | Nucleic acid | Sequences listed below in procedure; specific brand information will be left up to the discretion of the replicating lab and recorded later | | |
| *B2M*-primers (forward and reverse) | Nucleic acid | | | |
| 96-well multiplate PCR plates, clear | qPCR | Bio-Rad | MLL9601 | Original brand not specified |
| qPCR plate seals | Labware | Bio-Rad | MSB1001 | Or equivalent optically clear seals will be used |
| SYBR® green PCR master mix | qPCR | Life Technologies | 4344463 | – |
| DNA engine opticon system (qRT-PCR) | Instrument | Bio-Rad | n/a | Replaces ABI 7900 |
| Opticon monitor | Software | Bio-Rad | n/a | – |

## Procedure

### Note

- All cells will be sent for mycoplasma testing and STR profiling.
- MV4;11 and K-562 human leukemic cells maintained in RPMI-1640 medium, supplemented with 10% FBS and 1% penicillin/streptomycin at 37°C with 5% $CO_2$.

1. Seed MV4;11 or K-562 cells into 48-well tissue culture plates at $8 \times 10^4$ to $2 \times 10^5$ cells per well in triplicate. Incubate cells overnight at 37°C with 5% $CO_2$.
2. Treat cells with DMSO or 500 nM I-BET151 for 6 hr, in triplicate.
    a. Add drug directly to each well.
    b. Make stocks of I-BET151 at 1000× stock (500 µM) in DMSO.
    c. Final DMSO concentration kept to 0.1%.
3. Harvest cells and isolate RNA using the RNAspin mini kit according to manufacturer's instructions.
    a. Determine RNA purity ($A_{260/280}$ and $A_{260/230}$ ratios) and concentration.
4. Prepare cDNA using SuperScript III First-Strand Synthesis System according to the manufacturer's instructions.
5. Perform semi-quantitative PCR reactions, in triplicate, using *BCL2*-specific primers, *B2M*-specific primers (for normalization), and SYBR green PCR mastermix according to the manufacturer's instructions.
    a. Primers:
        i. *BCL2* forward: AGTACCTGAACCGGCACCT.
        ii. *BCL2* reverse: CAGCCAGGAGAAATCAAACAG.
        iii. *B2M* forward: TGACTTTGTCACAGCCCAAG.
        iv. *B2M* reverse: AGCAAGCAAGCAGAATTTGG.
6. Analyze data using the $\Delta\Delta C_T$ method: First, normalize *BCL2* values to *B2M* (housekeeping) values. Next, normalize I-BET151-treated cells to DMSO-treated cells to determine fold change of treatment relative to DMSO.
7. Repeat independently two additional times.

### Deliverables

- Data to be collected:

○ STR profile and result of mycoplasma testing of cells.
○ Purity ($A_{260/280}$ and $A_{260/230}$ ratios) and concentration of isolated total RNA from cells.
○ Raw qRT-PCR values, as well as analyzed $\Delta\Delta C_T$ values and bar graph of *BCL2* mRNA normalized to control mRNA levels for each condition. (Compare to Figure 3D).

## Confirmatory analysis plan

- Statistical analysis of the replication data:
  ○ Two-tailed *t*-tests with the Bonferroni correction:
    - Unpaired two-sample *t*-test of $\Delta\Delta C_T$ values from K562 cells compared to MV4;11 cells.
    - One-sample *t*-test of $\Delta\Delta C_T$ values from K562 cells compared to a constant of 1.
    - One-sample *t*-test of $\Delta\Delta C_T$ values from MV4;11 cells compared to a constant of 1.
- Meta-analysis of effect sizes:
  ○ Compute the effect sizes of each comparison, compare them against the effect size in the original paper and use a random effects meta-analytic approach to combine the original and replication effects, which will be presented as a forest plot.

## Known differences from the original study

All known differences are listed in the materials and reagents section above with the originally used item listed in the comments section. All differences have the same capabilities as the original and are not expected to alter the experimental design.

## Provisions for quality control

The cell lines used in this experiment will undergo STR profiling to confirm their identity and will be sent for mycoplasma testing to ensure there is no contamination. All of the raw data will be uploaded to the project page on the OSF (https://osf.io/hcqqy) and made publically available.

## Protocol 4: Assessment of maximum tolerable dose of I-BET151 in xenograft AML mouse model

This protocol assesses the maximum tolerable dose (MTD) of I-BET151 in a xenograft mouse model of leukemia by intra-peritoneal injection, using a range of I-BET151 compound. The original study reported using 30 mg/kg/day, however, as batch-to-batch variation occurs, the MTD will be assessed in this protocol to avoid toxicity. The MTD determined in this protocol will be used in Protocol 5 to assess the efficiency of I-BET151 in this model.

### Sampling

- Experiment has four cohorts:
  ○ Cohort 1: NOD/SCID mice treated daily with vehicle only.
  ○ Cohort 2: NOD/SCID mice treated daily with 10 mg/kg/day I-BET151.
  ○ Cohort 3: NOD/SCID mice treated daily with 20 mg/kg/day I-BET151.
  ○ Cohort 4: NOD/SCID mice treated daily with 30 mg/kg/day I-BET151.
- Experiment will use five mice per treatment group.
  ○ See 'Power calculations' section for details.

### Materials and reagents

| Reagent | Type | Manufacturer | Catalog # | Comments |
|---|---|---|---|---|
| MV4;11 | Human cell line | ATCC | CRL-9591 | – |
| I-BET151 (GSK1210151A) | Small molecule | Sigma–Aldrich | SML0666 | – |
| DMSO, molecular biology grade | Reagent | Sigma–Aldrich | D1435 | Original brand not specified |
| RPMI-1640 medium, with L-glutamine and sodium bicarbonate | Cell culture reagent | Gibco, Life Technologies | 22400-089 | Original catalog number not specified |
| Fetal bovine serum (FBS) | Cell culture reagent | Sigma–Aldrich | F2442 | Original brand not specified |

*Continued on next page*

*Continued*

| Reagent | Type | Manufacturer | Catalog # | Comments |
|---------|------|--------------|-----------|----------|
| Penicillin–streptomycin solution (100x) stabilized | Cell culture reagent | Invitrogen | 15140122 | Original brand not specified |
| Phosphate buffered saline (PBS) | Buffer | Gibco, Life Technologies | 14190-136 | – |
| Female and male NOD-SCID mice (6–8 weeks old) | Animal model | Jackson Laboratory | 001303 | – |
| IV Busulfex (busulfan) | Chemical | Otsuka America Pharmaceutical, Inc. | NDC 59148-070-90 | Not originally used |
| ½ cc LO-DOSE U-100 insulin syringe 28G | Labware | Becton–Dickinson | 329461 | Original brand not specified |
| APC anti-human HLA-A,B,C antibody | Antibodies | Biolegend | 311410 | Original catalog number not specified |
| APC mouse IgG2a, κisotype control antibody | Antibodies | Biolegend | 400220 | – |
| Kleptose HPB | Chemical | Roquette Pharma | n/a | Original brand not specified |
| 0.9% NaCl, USP | Chemical | Hospira, Inc | 0490-1966-05 | Original brand not specified |
| 1cc insulin syringe U-100 27G 5/8 | Labware | Becton–Dickinson | 329412 | Original brand not specified |
| Flow cytometer | Instrument | Becton–Dickinson | n/a | Canto or LSR II (replaces CyAn ADP from Dako) |
| FlowJo software | Software | Tree Star, Inc | n/a | – |

## Procedure

### Note

- All cells will be sent for mycoplasma testing and STR profiling, as well as screened against a Rodent Pathogen Panel.
- MV4;11 human leukemic cells maintained in RPMI-1640 medium, supplemented with 10% FBS and 1% penicillin/streptomycin at 37°C with 5% $CO_2$.

1. Receive non-obese diabetic/severely compromised immunodeficient (NOD-SCID) female and male mice (6–8 weeks old).
   a. An equal number of male and female mice should be used.
   b. Allow animals 1 week to acclimatize in a pathogen-free enclosure before start of study.
   c. Animals are housed in sterile conditions using high-efficiency particulate arrestance (HEPA)-filtered micro-isolator with 12-hr light/dark cycles, and fed with sterile rodent chow and acidified water ad libitum.
2. Condition mice with 30 mg/kg busulfan by intraperitoneal injection 24 hr prior to injection of MV4;11 cells.
3. Intravenously inject $1 \times 10^7$ MV4;11 cells in 0.2 ml sterile vehicle (PBS) into the tail vein of conditioned mice.
4. Monitor mice for engraftment:
   a. Inspect mice daily for signs of distress and record the scores using the 'Mouse Health Scoring System' (*Supplementary file 1*) for 21 days (*Cooke et al., 1996*).
   b. Weigh mice weekly for the entire duration of the experiment.
   c. At day 21 post-injection, collect retro-orbital bleeds and analyze leukemia burden (percent human HLA-A,B,C$^+$ cells) by flow cytometry.
      i. Stain samples with the following antibodies following manufacturer's recommendations:
         1. APC conjugated anti-human HLA-A,B,C.
         2. APC conjugated isotype control.
      ii. Perform flow cytometric analysis following manufacturer's instructions.
      iii. Gating strategy:
         1. On SSC vs FSC plot, gate on total nucleated population (both mouse and human cells).
         2. From the nucleated population, gate on HLA-A,B,C$^+$ cells (human leukemia cells).
5. Randomize mice into four cohorts using the following method:
   a. Exclude mice with no detectable leukemia burden.
      i. Use 0.5% human leukemia cells (HLA-A,B,C$^+$ cells) over the total live nucleated cells (human and mouse cells) in sample as a minimum threshold of engraftment (leukemia detected).

 b. Animals are randomized according to a stratified randomization procedure to balance gender and baseline tumor characteristics.
 i. Female and male mice are assigned into separate blocks.
 ii. In each block, animals are ranked according to disease burden (percent human HLA-A,B,C$^+$ cells) and group assignment is performed with a simple randomization procedure.

6. Begin once daily intraperitoneal injections with vehicle control, 10 mg/kg I-BET151, 20 mg/kg I-BET151, or 30 mg/kg I-BET151 (dose volume is 10 ml/kg).
 a. Prepare drug delivery vehicle: (10%) wt/vol, Kleptose HPB in 0.9%/g NaCl injection solution, pH 5.0.
 i. Weigh required amount of Kleptose HPB into a suitable glass container, for example, volumetric flask.
 ii. Make up to a final volume with 0.9%/g saline to achieve a 10% wt/vol, Kleptose HPB solution.
 iii. Mix contents until vehicle has visually clarified.
 b. Prepare initial formulation of I-BET151: 60 mg/ml of I-BET151 in DMSO (stock).
 i. Dispense the DMSO into the compounding vessel containing I-BET151.
 ii. Gently mix for minimum of 2 min or until complete dissolution achieved.
 c. Prepare the final drug formulation composition: 1, 2, or 3 mg/ml of I-BET151 in 5:95 vol/vol DMSO: drug delivery vehicle.
 i. Make a 20-fold dilution of the I-BET151 stock with the drug delivery vehicle; adjust pH to 5.0 using 2 M HCl to obtain a 3 mg/ml I-BET151 injection solution.
 1. Dispense the DMSO stock into a glass recipient vessel containing 25 ml of the required volume of vehicle.
 2. With the remaining 4.925 ml of vehicle, rinse the vessel containing the DMSO stock, adding the rinsed volume to the compounding vessel from step 1, removing any remaining dose by pipette.
 3. Gently mix for minimum of 1 min. A cloudy dose should form.
 4. Accurately add 2 µl of 2 M HCl to the compounding vessel by use of pipette.
 5. Gently mix for minimum of 1 min.
 6. Repeat steps 4 and 5 as required until a clear solution is formed.
 7. Take pH of resultant solution (final pH should be 5.0).
 ii. Make a 1.5-fold dilution of the 3 mg/ml I-BET151 injection solution with the drug delivery vehicle to obtain a 2 mg/ml I-BET151 injection solution.
 iii. Make a two-fold dilution of the 3 mg/ml I-BET151 injection solution with the drug delivery vehicle to obtain a 1 mgl/ml I-BET151 injection solution.
 d. Sub aliquot into fresh glass vials for use during the duration of the study.
 i. Make 21 aliquots for each injection solution (3 mg/ml, 2 mg/ml, and 1 mg/ml) at 1.8 ml/vial.
 ii. Store at 4°C.
 1. Dose stability has been determined for 21 days following storage at +4°C. If study duration is longer than 21 days another dose would need to be prepared on day 22.
 e. Bring one aliquot of each injection solution (3 mg/ml, 2 mg/ml, and 1 mg/ml) to room temperature before injection.

7. Continue dosing mice with either drug or vehicle every day for 21 days.
 a. Monitor mice daily for signs of disease (activity, posture, fur texture, and mobility).
 b. Weigh mice once a week.
 c. Record scores according to the 'Mouse Health Scoring System' (see step 4a).
 d. Euthanize mice when they receive a heath monitoring score of 3. This includes early signs of loss of hind limb motility, which is indicative of this disease model (*O'Farrell et al., 2003*; *Lopes de Menezes et al., 2005*).
 e. Euthanize all remaining mice within 3 days of the last treatment.

8. The MTD will be determined by identifying the dose at which the group body weight loss does not exceed 20% compared with the vehicle group and at which morbidity is not observed in one or more animals. When the MTD is reached, the next lowest dose will be used in Protocol 5.

## Deliverables

■ Data to be collected:
 ○ STR profile and result of mycoplasma and pathogen testing of cells.
 ○ Mouse health records (health monitoring [scores 0–3], weekly animal weights, date of treatment, euthanasia, and cause of termination).
 ○ Kaplan–Meier survival curves by group.

○ All flow cytometry plots in gating scheme (including controls), leading to final populations of HLA-A,B,C$^+$ cells before treatment intervention.

## Confirmatory analysis plan

- n/a.

## Known differences from the original study

The original study conditioned the recipient mice with a sublethal dose of radiation (300 cGy) prior to injection of MV4;11 cells. The replication attempt will use a single dose of busulfan, which has been reported to be comparable for human cell engraftment in NOD-SCID mice (*Robert-Richard et al., 2006*). All known differences are listed in the materials and reagents section above with the originally used item listed in the comments section. All differences have the same capabilities as the original and are not expected to alter the experimental design.

## Provisions for quality control

The cell lines used in this experiment will undergo STR profiling to confirm its identity and will be sent for mycoplasma testing to ensure there is no contamination. Additionally, cells used for xenograft injection will be screened against a Rodent Pathogen Panel to ensure no contamination prior to injection. The animals will be randomized prior to treatment. All of the raw data will be uploaded to the project page on the OSF (https://osf.io/hcqqy) and made publically available.

## Protocol 5: Generation of disseminated xenograft AML mouse model and testing of I-BET151 compound in vivo

This protocol assesses the efficacy of I-BET151 as a therapeutic agent in a xenograft mouse model of leukemia. Immunocompromised mice will be injected with preparations of MV4;11 cells and disease will progress for 21 days. At day 21, mice will be treated either with I-BET151 or vehicle control. Disease-free progression will be measured and plotted, as reported in Figure 4B. The presence and degree of disease progression will be determined by measuring the number of human leukemia cells present in the PB, spleen, and bone marrow (BM) of leukemic xenograft mice. Leukemic mice treated with I-BET151 will be compared to mice treated with vehicle control. This protocol replicates the experiments reported in Figure 4D and Supp. Figure 16A.

## Sampling

- Experiment has two cohorts:
  ○ NOD/SCID mice treated daily with vehicle only.
  ○ NOD/SCID mice treated daily with dose of I-BET151 determined in Protocol 4.
- Experiment will use 14 mice per treatment group.
  ○ To account for a higher censor rate, or exclusion, 14 mice will be used per group to ensure enough mice are included to reach a minimum power of 80%.
  ○ See 'Power calculations' section for details.

## Materials and reagents

| Reagent | Type | Manufacturer | Catalog # | Comments |
|---|---|---|---|---|
| MV4;11 | Human cell line | ATCC | CRL-9591 | – |
| I-BET151 (GSK1210151A) | Small molecule | Sigma–Aldrich | SML0666 | – |
| DMSO, molecular biology grade | Reagent | Sigma–Aldrich | D1435 | Original brand not specified |
| RPMI-1640 medium, with L-glutamine and sodium bicarbonate | Cell culture reagent | Gibco, Life Technologies | 22400-089 | Original catalog number not specified |
| Fetal bovine serum (FBS) | Cell culture reagent | Sigma–Aldrich | F2442 | Original brand not specified |
| Penicillin–streptomycin solution (100×) stabilized | Cell culture reagent | Invitrogen | 15140122 | Original brand not specified |
| Phosphate buffered saline (PBS) | Buffer | Gibco, Life Technologies | 14190-136 | – |

*Continued on next page*

*Continued*

| Reagent | Type | Manufacturer | Catalog # | Comments |
|---|---|---|---|---|
| Female and male NOD-SCID mice (6–8 weeks old) | Animal model | Jackson Laboratory | 001303 | – |
| IV Busulfex (busulfan) | Chemical | Otsuka America Pharmaceutical, Inc. | NDC 59148-070-90 | Not originally used |
| Ammonium chloride solution | Chemical | Stem Cell Technologies | 07850 | Replaces red blood cell lysis buffer from 5 prime |
| CountBright absolute counting beads | Flow cytometry reagent | Life Technology | C36950 | Not originally used |
| FACS lysing solution | Chemical | Becton–Dickinson | 349202 | Replaces red blood cell lysis buffer from 5 prime |
| ½ cc LO-DOSE U-100 insulin syringe 28G | Labware | Becton–Dickinson | 329461 | Original brand not specified |
| APC anti-human HLA-A,B,C antibody | Antibodies | Biolegend | 311410 | Original catalog number not specified |
| APC mouse IgG2a, κ isotype control antibody | Antibodies | Biolegend | 400220 | – |
| Annexin V-FITC Kit | Antibodies | Miltenyi Biotec Ltd | 130-092-052 | Original catalog number not specified |
| 7-AAD | Dye | BD Pharmingen | 51-68981E | Original catalog number not specified |
| Kleptose HPB | Chemical | Roquette Pharma | n/a | Original brand not specified |
| 0.9% NaCl, USP | Chemical | Hospira, Inc | 0490-1966-05 | Original brand not specified |
| 1cc insulin syringe U-100 27G 5/8 | Labware | Becton–Dickinson | 329412 | Original brand not specified |
| Flow cytometer | Instrument | Becton–Dickinson | n/a | Canto or LSR II (replaces CyAn ADP from Dako) |
| FlowJo software | Software | Tree Star, Inc | n/a | – |

## Procedure

### Note

- All cells will be sent for mycoplasma testing and STR profiling, as well as screened against a Rodent Pathogen Panel.
- MV4;11 human leukemic cells maintained in RPMI-1640 medium, supplemented with 10% FBS and 1% penicillin/streptomycin at 37°C with 5% $CO_2$.

1. Receive female and male NOD-SCID mice (6–8 weeks old).
   a. An equal number of male and female mice should be used.
   b. Allow animals 1 week to acclimatize in a pathogen-free enclosure before start of study.
   c. Animals are housed in sterile conditions using HEPA-filtered micro-isolator with 12-hr light/dark cycles, and fed with sterile rodent chow and acidified water ad libitum.
2. Condition mice with 30 mg/kg busulfan by intraperitoneal injection 24 hr prior to injection of MV4;11 cells.
3. Intravenously inject $1 \times 10^7$ MV4;11 cells in 0.2 ml sterile vehicle (PBS) into the tail vein of conditioned mice.
4. Monitor mice for engraftment as described in step 4 of Protocol 4.
5. Randomize mice into four cohorts using the following method.
   a. Exclude mice with no detectable leukemia burden.
      i. Use 0.5% human leukemia cells (HLA-A,B,C$^+$ cells) over the total live nucleated cells (human and mouse cells) in sample as a minimum threshold of engraftment (leukemia detected).
   b. Animals are randomized according to a stratified randomization procedure to balance gender and baseline tumor characteristics.
      i. Female and male mice are assigned into separate blocks.
      ii. In each block, animals are ranked according to disease burden (percent human HLA-A,B,C$^+$ cells) and group assignment is performed with a simple randomization procedure.

6. Begin once daily intraperitoneal injections with vehicle control or I-BET151 dose determined from Protocol 4 (dose volume is 10 ml/kg).
   a. Prepare vehicle and drug as outlined in step 6 of Protocol 4.
   b. The same lot of I-BET151 will be used.
7. Continue dosing mice with either drug or vehicle every day for 21 days.
   a. Monitor mice as described in step 7 of Protocol 4.
   b. Euthanize mice when they receive a health-monitoring score of 3 or within 3 days of the last treatment.
8. At sacrifice, collect PB by cardiac puncture into EDTA-treated tubes. Remove spleen and both tibias and femurs per mouse.
   a. Prepare cell suspensions from spleen (SPL) by pressing the spleen through a cell strainer in PBS and BM cells by flushing both tibias and femurs with PBS following the replicating lab's standard protocols.
   b. For HLA-A,B,C and apoptosis analysis (step 9 below), lyse red blood cells from samples using ammonium chloride solution following manufacturer's instructions.
   c. Collect two equal aliquots of cells for HLA-A,B,C and apoptosis analysis (step 9 below) and leukemia burden (step 10 below).
9. Perform flow cytometric analysis for apoptosis analysis in PB, SPL, and BM cells using Annexin V-FITC kit.
   a. Stain no more than $1 \times 10^6$ cells per sample with the following antibodies according to manufacturer's recommendations in PBS supplemented with 0.1% bovine serum albumin and 1 mM EDTA.
      i. APC conjugated anti-human HLA-A,B,C with 7-AAD and Annexin V-FITC.
      ii. APC conjugated isotype control with 7-AAD and Annexin V-FITC.
   b. Gating strategy:
      i. On FSC vs HLA-A,B,C plot, gate on HLA-A,B,C$^+$ cells (human leukemia cells).
      ii. From the leukemia cell population, use Annexin V vs 7-AAD plot to gate on the following cell populations:
         1. Annexin$^+$ 7-AAD$^-$ population (apoptotic cells).
         2. Annexin$^+$ 7-AAD$^+$ population (dead cells).
10. Perform flow cytometry analysis for leukemia burden in PB, SPL, and BM cells.
    a. Stain PB, SPL, and BM cells in a sample volume of 50 µl each. Add 20 µl of the following antibodies and incubate at room temperature for 15 min.
       i. APC conjugated anti-human HLA-A,B,C.
       ii. APC conjugated isotype control.
    b. Add CountBright absolute counting beads in 1× FACS lysing solution and incubate at room temperature for 15 min.
    c. Perform flow cytometric analysis following manufacturer's instructions.
    d. Gating strategy:
       i. On the FL3-H vs FSC plot, gate on CountBright absolute counting beads.
       ii. On SSC vs FSC plot, gate on total nucleated population (both mouse and human cells).
       iii. From the nucleated population, use HLA-A,B,C vs SSC plot to gate on HLA-A,B,C$^+$ cells (human leukemia cells).
11. For each mouse, confirm the presence or absence of leukemia. If a mouse is euthanized before the end of the experiment time length, but does not have detectable disease as assessed by flow cytometry, they should be censored from the Kaplan–Meier survival curve.
    a. Use 0.5% human leukemia cells (HLA-A,B,C$^+$ cells) over the total live nucleated cells (human and mouse cells) in sample as a minimum threshold of engraftment (leukemia detected).

## Deliverables

- Data to be collected:
  - STR profile and result of mycoplasma and pathogen testing of cells.
  - Mouse health records (health monitoring (scores 0–3), weekly animal weights, date of treatment, euthanasia, and cause of termination).
  - Kaplan–Meier survival curve comparing disease-free survival of I-BET151-treated xenografted mice vs vehicle-treated control xenografted mice. Compare to Figure 4B.
  - Include raw disease-free survival data for I-BET151 treated and untreated xenografted mice, including any mice censored because of no detectable disease.

○ All flow cytometry plots in gating scheme (including controls), leading to final populations of HLA-A,B,C$^+$ cells before and after treatment intervention. Compare to Figure 4D and Supplemental Figure S16A.
○ Number of HLA-A,B,C$^+$ cells in PB, SPL, and BM in each treatment group.

## Confirmatory analysis plan

■ Statistical analysis of the replication data:
○ Comparison of Kaplan–Meier survival curves between vehicle and I-BET151-treated mice using the Log-rank Mantel–Cox test.
■ Meta-analysis of effect sizes:
○ Compute the effect sizes of each comparison, compare them against the effect size in the original paper, and use a random effects meta-analytic approach to combine the original and replication effects, which will be presented as a forest plot.

## Known differences from the original study

The original study conditioned the recipient mice with a sublethal dose of radiation (300 cGy) prior to injection of MV4;11 cells. The replication attempt will use a single dose of busulfan, which has been reported to be comparable for human cell engraftment in NOD-SCID mice (*Robert-Richard et al., 2006*). The original study counted PB cells using a SciVet abc machine, while the replication attempt will include CountBright absolute counting beads to determine the absolute number of human leukemia cells in each mouse after treatment. The original study lysed red blood cells from samples using Red Blood Cell Lysis Buffer, while the replication attempt will use ammonium chloride solution while performing HLA-A,B,C and 7-AAD analysis. For analysis of leukemia burden using CountBright absolute counting beads, the cells will be lysed using 1× DB Lysis Buffer during manufacturer's instructions. All known differences are listed in the materials and reagents section above with the originally used item listed in the comments section. All differences have the same capabilities as the original and are not expected to alter the experimental design.

## Provisions for quality control

The cell lines used in this experiment will undergo STR profiling to confirm their identity and will be sent for mycoplasma testing to ensure there is no contamination. Additionally, cells used for xenograft injection will be screened against a Rodent Pathogen Panel to ensure no contamination prior to injection. The animals will be randomized prior to treatment. The apoptotic marker dye 7-AAD will be used to exclude populations of dead or dying cells from analysis and an isotype control antibody will be used to confirm the specificity of the HLA-A,B,C antibody. All of the raw data will be uploaded to the project page on the OSF (https://osf.io/hcqqy) and made publically available.

# Power calculations

For additional details on power calculations, please see analysis scripts and associated files on the Open Science Framework:
https://osf.io/bdk6c/.

# Protocol 1

Not applicable.

# Protocol 2

Summary of original data reported in Figures 2A, S11A, and S11B:

| Cell line | IC$_{50}$ |
|---|---|
| K562 cells (−MLL) | >100 µM |
| MV4;11 cells (+MLL) | 26 nM |

The original data do not indicate the error associated with multiple biological replicates. To identify a suitable sample size, power calculations were performed using different levels of relative variance.

## Test family

■ Two-tailed *t*-test, difference between two independent mean values, alpha error = 0.05.

'Power calculations' performed with G*Power software, version 3.1.7 (*Faul et al., 2007*).

| Group 1 | Group 2 | Effect size *d* | A priori power | Group 1 sample size | Group 2 sample size |
|---|---|---|---|---|---|
| 2% variance | | | | | |
| K562 | MV4;11 | 70.69229 | 99.9% | 2 | 2 |
| 15% variance | | | | | |
| K562 | MV4;11 | 9.42564 | 98.8 | 2 | 2 |
| 28% variance | | | | | |
| K562 | MV4;11 | 5.04945 | 99.4% | 3 | 3 |
| 40% variance | | | | | |
| K562 | MV4;11 | 3.53461 | 89.2% | 3 | 3 |

In order to produce quantitative replication data, we will run the experiment three times. Each time we will determine the $IC_{50}$. The three replicates and the original reported value will be checked to see if the original value is an outlier using Grubb's test (with a significance level of 0.05). If the original value is detected as an outlier it will not be included with the replication replicates to determine the standard deviation of $IC_{50}$ values, otherwise it will be included in the standard deviation calculation. The calculated standard deviation will be combined with the reported value from the original study to simulate the original effect size. We will use this simulated effect size to determine the number of replicates necessary to reach a power of at least 80%. We will then perform additional replicates, if required, to ensure that the experiment has more than 80% power to detect the original effect.

## Protocol 3

Summary of original data estimated from graph reported in Figure 3D:

| Cell line | Treatment | Mean | Stdev | N |
|---|---|---|---|---|
| K562 cells (−MLL) | DMSO | 1 | 0 | 3 |
| | I-BET151 | 0.22 | 0.03 | 3 |
| MV4;11 cells (+MLL) | DMSO | 1 | 0 | 3 |
| | I-BET151 | 0.935 | 0.05 | 3 |

## Test family

■ Two-tailed *t*-test, difference between two independent mean values, Bonferroni's correction: alpha error = 0.01667.

'Power calculations' performed with G*Power software, version 3.1.7 (*Faul et al., 2007*).

| Group 1 | Group 2 | Effect size *d* | A priori power | Group 1 sample size | Group 2 sample size |
|---|---|---|---|---|---|
| MV4;11, I-BET151 treated | K562, I-BET151 treated | 17.34130 | 99.3%* | 2* | 2* |

*Three samples per group will be used based on the other planned tests making the power 99.9%.

## Test family

- Two-tailed *t*-test, difference from constant (one sample case), Bonferroni's correction: alpha error = 0.01667.

  'Power calculations' performed with G*Power software, version 3.1.7 (*Faul et al., 2007*).

| Group 1 | Group 2 | Effect size *d* | A priori power | Group 1 sample size | Group 2 sample size |
|---------|---------|------|------|------|------|
| MV4;11, DMSO treated | MV4;11, I-BET151 treated | 26.000 | 99.9% | 3 | 3 |

## Test family

- Two-tailed *t*-test, difference from constant (one sample case), Bonferroni's correction: alpha error = 0.01667.

  Sensitivity calculations performed with G*Power software, version 3.1.7 (*Faul et al., 2007*).

| Group 1 | Group 2 | Detectable effect size *d* | A priori power | Group 1 sample size | Group 2 sample size |
|---------|---------|------|------|------|------|
| K562, DMSO treated | K562, I-BET151 treated | 5.66748* | 80.0% | 3* | 3* |

*Since the original comparison was not statistically significant. This is the effect size that can be detected with 80% power and the indicated sample size. Unlike the above power calculations, the aim of this sensitivity calculation is not to detect the original effect size, but to understand what effect size could be detected. The original effect size is 1.300.

## Protocol 4

The law of diminishing return was used to determine the sample size for assessment of MTD (*Charan and Kantharia, 2013*).

- E = Total number of animals − Total number of groups.

| Number of treatment groups | Total sample size with E = 10 | Total sample size with E = 20 |
|------|------|------|
| 4 | 14* | 24* |

*To keep animals per group balanced 16 (4 per group), 20 (5 per group), or 24 (6 per group) total samples keeps E between 10 and 20. 20 total animals (5 per group) will be used to account for any potential exclusion.

## Protocol 5

Summary of original data estimated from Kaplan–Meier graph reported in Figure 4B:

| Treatment group | Median survival | Hazard ratio (to vehicle control) | N | Censoring rate (# censored/day) |
|------|------|------|---|------|
| Vehicle-treated mice | 14 days | N/A | 5 | 0 |
| I-BET151-treated mice | N/A | 0.09687 | 5 | 0.09524* |

*Two mice were censored from the I-BET151 cohort during the 21-day treatment period. For the power calculations, the censoring rate was divided in half since the calculation assumes the censoring rate is equal for both groups.

## Test family

- Log-rank (Mantel–Cox) test: alpha error = 0.05.

'Power calculations' performed with the Sample Size Calculator hosted by the UCSF Clinical and Translational Science Institute (*Schoenfeld, 1983*).

| Group 1 | Group 2 | Treatment duration | A priori power | Group 1 total events needed | Group 1 sample size | Group 2 total events needed | Group 2 sample size |
|---------|---------|--------------------|----------------|------------------------------|---------------------|------------------------------|---------------------|
| Vehicle | I-BET151 | 21 days | 80.0% | 5* | 12* | 1* | 12* |

*14 per group will be used to account for a potential higher censoring rate, or exclusion.

## Acknowledgements

The Reproducibility Project: Cancer Biology core team would like to thank the original authors, in particular Mark Dawson for generously sharing critical information to ensure the fidelity and quality of this replication attempt. We thank Courtney Soderberg at the Center for Open Science for assistance with statistical analyses. We would also like to thank the following companies for generously donating reagents to the Reproducibility Project: Cancer Biology; American Type Culture Collection (ATCC), Applied Biological Materials, BioLegend, Charles River Laboratories, Corning Incorporated, DDC Medical, EMD Millipore, Harlan Laboratories, LI-COR Biosciences, Mirus Bio, Novus Biologicals, Sigma–Aldrich, and System Biosciences (SBI).

## Additional information

### Group author details

**Reproducibility Project: Cancer Biology**

Elizabeth Iorns: Science Exchange, Palo Alto, California; William Gunn: Mendeley, London, United Kingdom; Fraser Tan: Science Exchange, Palo Alto, California; Joelle Lomax: Science Exchange, Palo Alto, California; Nicole Perfito: Science Exchange, Palo Alto, California; Timothy Errington: Center for Open Science, Charlottesville, Virginia

### Competing interests

JJF: ProNovus Bioscience is a Science Exchange associated laboratory. AK: ProNovus Bioscience is a Science Exchange associated laboratory. XS: Stem Cell and Xenograft Core, University of Pennsylvania Perelman School of Medicine is a Science Exchange associated laboratory. GD-D: Stem Cell and Xenograft Core, University of Pennsylvania Perelman School of Medicine is a Science Exchange associated laboratory. RP:CB: EI, FT, JL, and NP: Employed by and hold shares in Science Exchange Inc. The other authors declare that no competing interests exist.

### Funding

| Funder | Author |
|--------|--------|
| Laura and John Arnold Foundation | Reproducibility Project: Cancer Biology |

The Reproducibility Project: Cancer Biology is funded by the Laura and John Arnold Foundation, provided to the Center for Open Science in collaboration with Science Exchange. The funder had no role in study design or the decision to submit the work for publication.

### Author contributions

JJF, AK, XS, GD-D, MG, KO, Drafting or revising the article; RP:CB, Conception and design, Drafting or revising the article

## Additional files

### Supplementary file

• Supplementary file 1. Mouse Health Scoring System.

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
