## [Decision Letter]

Thank you for submitting your work entitled “Registered report: Inhibition of BET recruitment to chromatin as an effective treatment for MLL-fusion leukaemia” for peer review at *eLife*. Your submission has been favourably evaluated by Charles Sawyers (Senior Editor), a Reviewing Editor, and three reviewers.

The reviewers have discussed their reviews with one another and the Reviewing Editor has drafted this decision to help you prepare a revised submission.

Summary:

The manuscript is designed to replicate the major findings of Dawson et al.,‘Inhibition of BET recruitment to chromatin as an effective treatment for MLL-fusion leukaemia’ (Nature, 2011).

There are 3 main experiments chosen for replication. (i) To replicate the sensitivity of MLL rearranged leukaemia cell lines to BET bromodomain inhibition (ii) To assess the downregulation of BCL2, a major target gene and (iii) To assess the efficacy of I-BET against a xenograft model of leukaemia.

Detailed protocols are provided for replicating each of these key experiments, with increased sample sizes above the original manuscript to ensure statistical significance of results.

Essential revisions:

1) In general, the authors summarize the literature adequately. This area of biomedical/cancer research has been very active and there have been many studies that have demonstrated pre-clinical efficacy for BET bromodomain inhibition in various malignancies. Nonetheless the authors should cite Zuber et al. (Nature, 2011); Delmore et al. (Cell, 2011) and Mertz et al. (PNAS, 2011) as these studies were published concurrently to the Dawson et al. study. In particular:

a) The efficacy of JQ1 (an independent BET inhibitor) in MV4-11 xenografts was shown in Figure 5D of this study: Mertz JA et al. (Proc Natl Acad Sci, 2011, Oct 4;108(40):16669-74).

b) The differential sensitivity of MV4-11 and K562 proliferation in vitro was demonstrated in Supplement Figure 6 of this study: Zuber J et al. (Nature, 2011, Aug 3;478(7370):524-8).

2) From a statistical perspective, the proposal looks very detailed and acknowledges managing some of the uncertainty in the study design. There is also clear discussion of the randomisation processes to be used. However, there are some aspects that necessitate some clarification.

a) Protocol 2 allows for the fact that the original report does not report any error for the average value reported. Was any attempt made to get this data from the original authors? Getting the original data for error would be preferable to generating replicates to get an estimate of the SD from 3 new biological replicates and the original report. If that is not possible then I think the 3 new biological replicates will be added to the original report to make a sample size of 4 to estimate the SD. Then this SD will be used to scale the original (singly reported) value as a scaled effect size and then a new sample size calculation will be performed to see how many more than 3 will be needed to achieve 80% power.

b) Protocol 3. Within this protocol there is a sample size calculation (sensitivity) to see what effect size could be detected with n=3. However, this comparison was not statistically significant in the original report. I think the aim of this part of the protocol should be reworded.

c) Protocol 4. Five mice per group will be used in the MTD analysis and this will be reported using Kaplan Meier plots. However, no sample size justification was made. What is the reasoning for using 5 per group?

---

## [Author Response]

*1) In general, the authors summarize the literature adequately. This area of biomedical/cancer research has been very active and there have been many studies that have demonstrated pre-clinical efficacy for BET bromodomain inhibition in various malignancies. Nonetheless the authors should cite Zuber et al. (Nature, 2011); Delmore et al. (Cell, 2011) and Mertz et al. (PNAS, 2011) as these studies were published concurrently to the Dawson et al. study. In particular*:

*a) The efficacy of JQ1 (an independent BET inhibitor) in MV4-11 xenografts was shown in Figure 5D of this study: Mertz JA et al. (Proc Natl Acad Sci, 2011, Oct 4;108(40):16669-74)*.

*b) The differential sensitivity of MV4-11 and K562 proliferation in vitro was demonstrated in Supplement Figure 6 of this study: Zuber J et al. (Nature, 2011, Aug 3;478(7370):524-8)*.

Thank you for bringing these to our attention. We have updated the manuscript to include these references.

*2) From a statistical perspective, the proposal looks very detailed and acknowledges managing some of the uncertainty in the study design. There is also clear discussion of the randomisation processes to be used. However, there are some aspects that necessitate some clarification*.

*a) Protocol 2 allows for the fact that the original report does not report any error for the average value reported. Was any attempt made to get this data from the original authors? Getting the original data for error would be preferable to generating replicates to get an estimate of the SD from 3 new biological replicates and the original report. If that is not possible then I think the 3 new biological replicates will be added to the original report to make a sample size of 4 to estimate the SD. Then this SD will be used to scale the original (singly reported) value as a scaled effect size and then a new sample size calculation will be performed to see how many more than 3 will be needed to achieve 80% power*.

We did reach out to the original authors about obtaining the originally reported data. Unfortunately, they were unable to provide the raw data for any of the included experiments. Thank you for the suggestion regarding the approach to used the replication variation and the original (singly reported) value to calculate the effect size to use in a new sample size calculation. The one potential issue that could potentially arise is if the replication data are significantly different than the original value, which would cause the SD to greatly increase. Thus, we propose to include the original value as long as it is not detected as an outlier using Grubb’s test. We have updated the manuscript to include this approach.

*b) Protocol 3. Within this protocol there is a sample size calculation (sensitivity) to see what effect size could be detected with n=3. However, this comparison was not statistically significant in the original report. I think the aim of this part of the protocol should be reworded*.

Thank you for the suggestion. We have reworded this part to increase the clarity of the calculation.

c) Protocol 4. Five mice per group will be used in the MTD analysis and this will be reported using Kaplan Meier plots. However, no sample size justification was made. What is the reasoning for using 5 per group

We have included the justification in the revised manuscript. The MTD will be determined using body weight loss and morbidity, while also reporting overall survival using Kaplan Meier plots. Sample size was determined using the law of diminishing return (since the primary aim is to find any level of difference between groups), with 5 animals per group sufficient to keep E between 10 and 20, while also accounting for any potential loss due to exclusion.